# Randomised controlled trial of gradual antipsychotic reduction and discontinuation in people with schizophrenia and related disorders: the RADAR trial (Research into Antipsychotic Discontinuation and Reduction)

Joanna Moncrieff,[1] Glyn Lewis,[2] Nick Freemantle,[3] Sonia Johnson [ID],[2] Thomas R E Barnes,[4] Nicola Morant [ID],[2] Vanessa Pinfold,[5] Rachael Hunter [ID],[6] Lyn J Kent,[7] Ruth Smith,[8] Katherine Darton,[9] Robert Horne,[10] Nadia E Crellin,[2] Ruth E Cooper,[11] Louise Marston,[6] Stefan Priebe[11]

For numbered affiliations see end of article.

**Correspondence to**
Dr Joanna Moncrieff, Mental Health Sciences, University College London and North East London mental health trust, London, UK; j.moncrieff@ucl.ac.uk

## ABSTRACT

**Introduction** Antipsychotic medication is effective in reducing acute symptoms of psychosis, but it has a range of potentially serious and debilitating adverse effects and is often disliked by patients. It is therefore essential it is only used when benefits outweigh harms. Although multiple trials conducted with people with schizophrenia indicate an increased risk of relapse in the short-term following abrupt antipsychotic discontinuation, there is little evidence about the long-term outcome of a gradual process of reduction and discontinuation on social functioning, relapse and other outcomes.

**Methods and analysis** This is a multicentre, randomised controlled trial involving people with schizophrenia and related disorders who have had more than one episode. Participants are randomised to have a clinically-supervised, gradual reduction of antipsychotic medication, leading to discontinuation when possible, or to continue with maintenance treatment. Blinded follow-up assessments are conducted at 6, 12 and 24 months and the primary outcome is social functioning, measured by the Social Functioning Scale at 24 months. A minimum of 134 evaluable participants provides 90% power to detect a five-point difference, and 206 to detect a four-point difference. Secondary outcomes include severe relapse (admission to hospital) and the study is also intended to detect a minimum 10% difference in severe relapse, which requires 402 participants, assuming a 15% loss to follow-up. Other secondary outcomes include all relapses, as identified by an independent and blinded endpoint committee, symptoms measured by the Positive and Negative Syndrome Scale, quality of life, adverse effects, self-rated recovery and neuropsychological measures. Enrolment started in 2016. The trial is scheduled to finish in June 2022.

### Strengths and limitations of this study

► The trial will be one of the first to provide data on the outcome of a gradual process of antipsychotic reduction and discontinuation in people with schizophrenia and related disorders.
► The trial will provide relatively long-term outcome data on social functioning as well as relapse and other outcomes.
► There are likely to be some deviations from the planned treatment strategies.
► Longer-term follow-up would be desirable and will be initiated in the future.

**Ethics and dissemination** Ethical approval was initially obtained on 27 October 2016 (UK Research Ethics Committee reference 16/LO/1507). Results will be published in peer-reviewed journals and disseminated to the public.

**Trial registration number** ISRCTN90298520. EudraCT: 2016-000709-36. Pre-results.

## INTRODUCTION

Schizophrenia and related conditions affect up to 1% of the population,[1] and are associated with long-term suffering and disability, premature death, physical illness and high costs to individuals and society.[2] Recommended treatment for people with recurrent episodes consists of continuing antipsychotic medication.[3 4] Current guidelines do not recommend attempts at reduction or discontinuation of antipsychotics after the first episode, and in practice, antipsychotic

BMJ

treatment is often 'for life'. In spite of ongoing treatment, many people remain functionally impaired. In one study 25% of people with schizophrenia had severe social disabilities after 15 years, and only 14% had none.[5]

The evidence base for long-term antipsychotic treatment consists of studies showing lower relapse rates with maintenance treatment compared with discontinuation.[6] However, there are acknowledged methodological problems with these studies.[7] In addition, several long-term non-randomised cohorts studies find worse outcomes in people who take continuous treatment compared with those who do not, although confounding by indication is likely to be relevant.[8 9] Criticism of randomised controlled trials includes the fact that most focus on relapse and neglect other outcomes. Follow-up is generally short; only six of 65 studies included in a recent meta-analysis by Leucht *et al*[6] had followed the participants up for more than a year, and longer duration of follow-up was found to be associated with less difference between maintenance and discontinuation.[6 10] Moreover, relapse rates may have been inflated by abrupt discontinuation and misidentification of withdrawal-related adverse effects.[11 12] Some evidence suggests gradual discontinuation may reduce risk of relapse compared with abrupt discontinuation,[13] although this difference was not confirmed in the analysis by Leucht *et al*.[6] However, the average taper of 28 days for gradual medication withdrawal may not have prevented discontinuation-related effects in those people who had been taking antipsychotics for many years.

Long-term antipsychotic medication is associated with potentially serious physical complications, including diabetes, tardive dyskinesia and cardiovascular disease.[14–17] Other adverse effects, such as sexual dysfunction, sedation, emotional blunting and akathisia may be debilitating and unpleasant.[18–20] Previous research has found that many patients find the adverse effects of antipsychotics burdensome, and would like to try and discontinue the treatment at some stage.[21–23]

There are few long-term follow-ups of people from randomised studies of antipsychotic discontinuation. A 7 year follow-up of an 18 month open trial conducted with people with first episode psychosis found that assignment to a gradual antipsychotic reduction programme was associated with better rates of social recovery and equal rates of relapse compared with maintenance treatment.[24] Improved levels of neurocognitive performance exhibited by those randomised to antipsychotic reduction at 18 months suggest a possible mechanism for the long-term differences in functional outcome.[25] In contrast, recent data from a 10 year follow-up of a 1 year, placebo-controlled trial of quetiapine in people with first-episode psychosis reported higher rates of a composite 'poor outcome' in people originally randomised to placebo. There were no differences, however, in overall symptom measures, social functioning or quality of life.[26]

Therefore, although continuing antipsychotic treatment has become the norm, it remains unclear whether it has an optimal risk-benefit balance for all people with psychosis or schizophrenia. In particular, more evidence on the effects of gradual reduction of antipsychotics on short and longer-term outcomes could inform practice in this area. Effects on social functioning are particularly important to study because of evidence that long-term antipsychotic treatment may impair social functioning, despite improving symptoms or reducing relapse in the short-term.[24 27]

Further studies are currently being conducted in the first-episode psychosis population.[28] To date, however, there is no study using gradual and flexible reduction with long-term follow-up in people with more than one episode.

### Aims of trial

The current trial was designed to compare the benefits and harms of a gradual programme of dose reduction and discontinuation of antipsychotic treatment, under clinician guidance, with maintenance treatment in people with a diagnosis of schizophrenia or a related disorder. In particular, the study was set up to test whether such a strategy can improve functional outcomes in people with recurrent or chronic psychosis while minimising the risk of worsening symptoms or relapse.

## METHODS AND ANALYSIS

### Study design

The trial consists of an open, parallel group randomised trial with concealed, individual randomisation. Randomisation is conducted through an independent, internet-based system linked with the database (https://www.sealedenvelope.com/randomisation/internet/) with 1:1 allocation. There are no replacements for participants who drop out or otherwise cannot comply with study procedures.

Participants and clinicians are aware of allocation, but outcome assessors are maintained blind to intervention arm as far as possible. This is achieved by nominating separate blinded and unblinded researchers at each site, and ensuring that blinded researchers are not exposed to discussions or written information that would reveal allocation. Following follow-up assessment blinded research staff record if they suspect they have guessed arm allocation.

The trial lasts for 2 years, with follow-up assessments conducted at 6, 12 and 24 months post randomisation. The trial design is presented in figure 1. Ethical approval includes the possibility of conducting a longer-term follow-up, and resources will be sought to support this at a later date.

### Interventions

The antipsychotic reduction protocol was developed after consultation with professionals, academics and the trial's RADAR Lived Experience Advisory Panel (LEAP). Both the antipsychotic reduction and maintenance protocols are administered by treating psychiatrists. For those

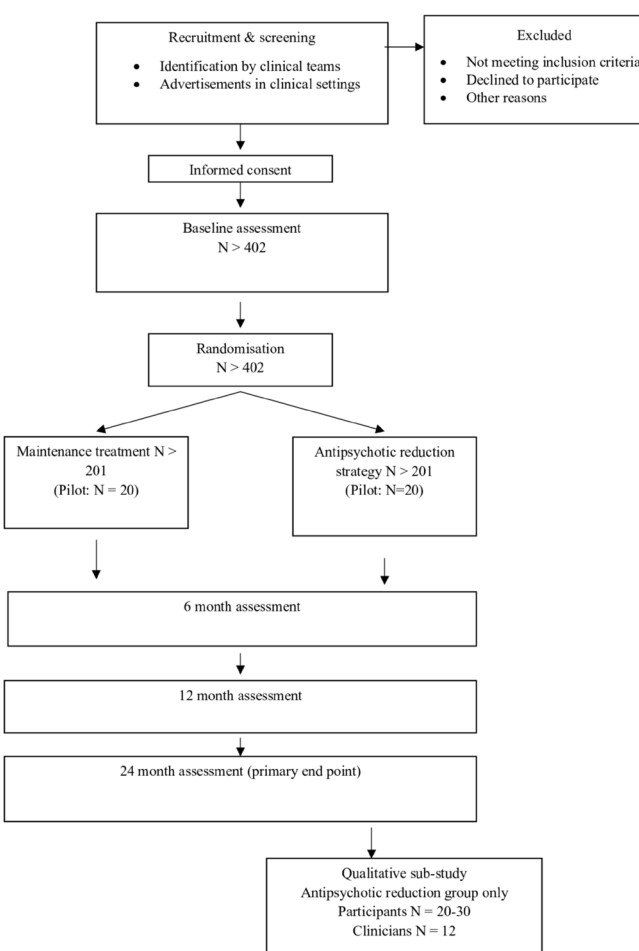

**Figure 1** Trial flow chart.

participants randomised to the antipsychotic medication reduction arm, an individualised reduction schedule is devised by the research team for each participant based on clinical judgement and adjusted according to the participant's initial antipsychotic regimen. The dose is reduced incrementally every 1 or 2 months, focusing on one drug at a time where participants are taking more than one antipsychotic. The rate of reduction varies according to baseline dose, with most schedules aiming for discontinuation within 12 months, but some lasting longer where baseline doses are high. Treating psychiatrists are asked to see the participants who have been randomised to antipsychotic reduction approximately every 2 months for the duration of the reduction, to adjust the medication regimen and monitor mental state. The participants are offered the option to discontinue antipsychotic medication completely if the reduction progresses well, or to reduce to a very low dose, defined as the equivalent of 2 mg of haloperidol a day or less, which is lower than the minimum recommended therapeutic dose for most antipsychotics. This is twice the dose that was defined as a 'low dose' in the Dutch first episode study,[24] since some research suggests that doses of antipsychotic medication in people with first episodes should be around half those recommended for people with more established

disorders.[29 30] Guidance on the antipsychotic reduction strategy stresses the need for flexibility, and includes a suggested protocol for the treatment of adverse reactions to withdrawal or symptom exacerbation. Participants randomised to maintenance treatment are requested not to make major reductions in their dose of antipsychotic medication during the trial period. Increases in dose are permitted within the protocol, as are changes to a different antipsychotic agent at the same equivalent dose and minor dose reductions to address side effects.

Participants are monitored by their care team according to usual practice. Patient records are scrutinised by members of the research team every 2 months to monitor the progress of the antipsychotic reduction and adherence to the maintenance protocol. Any deviations, such as participants allocated to reduction not starting or halting their reduction, are discussed with the treating clinician. All changes in antipsychotic doses and use of other medications is recorded throughout the study.

Other drug treatment and interventions such as psychological therapies may be used as indicated throughout the trial.

### Inclusion and exclusion criteria
The eligibility criteria are detailed in box 1. They are designed to make the trial as generalisable as possible to routine clinical practice, while minimising the risks of antipsychotic reduction in those with a history of posing a serious risk to self or others.

### Outcome measures
The primary outcome is social functioning at 2 year follow-up. This will be measured using the Social Functioning Scale (SFS). The scale was developed in 1990,

---

**Box 1    Eligibility criteria**

**Inclusion criteria**
1. Aged 18 years or older.
2. A clinical and/or International Classification of Diseases, 10th Revision, diagnosis of schizophrenia, schizoaffective disorder, delusional disorder or other non-affective psychosis.
3. More than one previous episode or psychotic exacerbation, or a single episode lasting more than 1 year.
4. Prescribed continuing antipsychotic medication.

**Exclusion criteria**
1. Lack of capacity to consent to the trial.
2. Insufficient command of spoken English to understand trial procedures.
3. Subject to a section of the Mental Health Act that includes a requirement to take antipsychotic medication.
4. Clinician considers there will be a serious risk of harm to self or others.
5. Admitted to hospital or treated by a Home Treatment or Crisis Team within the last month.
6. Women who have a confirmed pregnancy.
7. Women who are breastfeeding.
8. Involvement in another Investigational Medicinal Product (IMP) trial.

---

and has good reliability.[31 32] It can demonstrate change over time[33–35] and distinguishes different types of antipsychotic treatment.[33–36] It also distinguishes between people in remission and those with an ongoing episode,[37] people with long versus short duration of untreated psychosis,[38] people who convert to psychosis versus those who do not in an 'ultra-high risk' cohort[34] and between people who are employed versus unemployed.[31] This scale was preferred to the Groningen Social Disability Scale (GSDS),[39] which was used in the Dutch first-episode study, because of the GSDS is long, requires lengthy training and has a strong focus on relationships rather than functioning.

The main measure of the safety of the trial is severe relapse which is defined as admission to acute psychiatric inpatient treatment. This was selected because it is an objective criterion, can be precisely dated and can be obtained through patient records. Previous antipsychotic withdrawal studies have defined relapse in numerous different ways, with no consensus on defining criteria. Therefore, the current trial will also use an expert endpoint committee to assess the presence or absence of relapse based on blinded information from clinical case notes, based on predefined criteria and guidance.

Other outcome measures include: symptoms as measured by the Positive and Negative Syndrome Scale,[40] subjective quality of life as measured by the Manchester Short Assessment of quality of life,[41] adverse effects of antipsychotics measured by a modified version of the Glasgow Antipsychotic Side-effect Scale,[42] body weight, sexual dysfunction as measured by the Arizona Sexual Experiences Scale,[43] Work Productivity and Activity Impairment - General Health (WPAI) questionnaire,[44] neuropsychological function (measured by a brief battery of tests designed for this trial, recovery as measured by the Questionnaire about the Process of Recovery,[45] ICECAP-A (quality of life) and EQ-5D-5L (quality of life).

In addition, process data will be collected, including dose of antipsychotic medication (in haloperidol equivalents), patient satisfaction as measured by the Client Satisfaction Questionnaire[46] and antipsychotic medication adherence as measured by Medication Adherence Rating Scale.[47]

Table 1 shows the time points at which each measure will be completed. All outcome measures are administered by research staff who have received standardised training. Data collected during assessments is stored in a secure online data management system provided by the Sponsor. Participant retention is promoted by sending correspondence and study newsletters to participants to facilitate engagement throughout the trial duration.

**Table 1** Time points at which different outcomes will be assessed

| Visit number | Baseline 1. | Follow-up 2. Pilot trial 3 month data collection | 3. 6 month follow-up | 4. 12 month follow-up | 5. 24 month follow-up | 6. Qualitative evaluation |
|---|---|---|---|---|---|---|
| Informed consent | X | | | | | X |
| Eligibility determination | X | | | | | |
| Protocol assessments | A-I, L-O, Q, R | K | C-G, I-O, Q, R | C-O,Q, R | C-R | Indicative topic guide with sample of participants and psychiatrists |
| Randomisation | X | | | | | |
| IMP administration | X | X | X | X | X | |
| Adverse events review | X | X | X | X | X | |
| Medical notes review for prescribing information and fidelity to intervention protocols | X | X | X | X | X | |
| Concomitant medication review | X | X | X | X | X | |

A, Demographic information (selected sections including weight and use of illicit drugs and alcohol); B, Diagnosis (established from clinical records); C, Social Functioning Scale (SFS); D, Positive and Negative Syndrome Scale (PANSS); E, Glasgow Antipsychotic Side-effect Scale (GASS); F, Client Satisfaction Questionnaire (CSQ-8); G, Manchester Short Assessment of quality of life (MANSA); H, Neuropsychological function tests; I, Medication Adherence Rating Scale (MARS-5); J, Relapse questionnaire; K, Serious Adverse Events; L, EQ-5D-5L; M, ICECAP-A; N, Client Service Receipt Inventory; O, Work Productivity and Activity Questionnaire; P, Schedule for economic data from patient records; Q, Questionnaire about the Process of Recovery (QPR); R, Arizona Sexual Experiences Scale (ASEX).
IMP, Investigational Medicinal Product.

## Recruitment

Participants are recruited from a variety of clinical teams within mental health services across the UK. Potential participants are identified initially by clinical staff or recruited by advertisements placed in clinical settings. Those who agree are sent further information about the study and then a baseline assessment is arranged after further discussion. Assessments are conducted at the patient's home or in clinical premises according to patient preference.

## Risk reduction

The eligibility criteria are designed to exclude people with known high risks of causing harm to themselves or other people. In addition, the gradual nature of the antipsychotic reduction will enable detection and treatment of early signs of relapse. All participants will receive usual care and monitoring of their mental state and behaviour by their clinical team. Those randomised to antipsychotic reduction will have increased contact with a psychiatrist for the duration of the reduction, mirroring usual clinical practice with someone undergoing a significant reduction of medication.

## Qualitative assessment

There will be a qualitative substudy involving around 20 to 30 participants from the antipsychotic reduction group and around 12 clinicians with experience of delivering the antipsychotic reduction programme. This study has two aims: to collect data on participant and clinician experience of trial processes, and to explore in detail experiences of antipsychotic reduction and discontinuation from the patient perspective. Participants will be identified towards the end of follow-up, using purposive sampling to obtain variation in clinical profile, experience of reduction, completion of the antipsychotic reduction protocol and experience of relapse.

Participants will be interviewed using a semi-structured interview schedule developed in collaboration with the RADAR LEAP. This will explore patients' experiences of the intervention and its impact on their mental health and wider lives, the acceptability of antipsychotic reduction, satisfaction with available support and responses to adverse events. Interviews will be conducted with early participants following the 24 month follow-up interviews so as not to confound the main trial results. Interviews with clinicians will explore their experiences of implementing different aspects of the antipsychotic reduction strategy, relations with the research team, acceptability of medication monitoring procedures and responses to adverse events. The data will be analysed using thematic analysis.

## Statistical analysis

The primary outcome (SFS) will be analysed using generalised mixed models, accounting for baseline and treatment periods. The principal analysis will be undertaken using the intention to treat population, including all

available data, without imputation. Supportive analyses will include analysis of different time periods, analysis using repeated measurements (where all subjects with data at one or more post intervention assessment will be included), exploring potential centre effects and making pessimistic assumptions by group on the patient outcome (to assess the thresholds to the impact of 'missingness') because of the likely pattern of data missing not at random.

The proportions of patients who experience a 'severe relapse' (psychiatric hospital admission) will be compared with survival analysis using Cox constant proportional hazards models and also between randomised groups at the two-year follow-up point. We will undertake supportive threshold analyses where any loss to follow-up will be considered according to extreme data patterns — for example, assuming all those who were lost in the antipsychotic withdrawal group had a relapse and all those in the maintenance group did not, and thus quantifying the extent of possible effects of missingness and thus the robustness of the main result.

A full statistical analysis plan will be completed and agreed prior to any analysis or unblinding of the data, including planned subgroup analyses and with detailed description of the statistical processes to be used.

## Economic analysis

The economic evaluation will evaluate the cost-effectiveness of the antipsychotic reduction strategy compared with maintenance treatment over 24 months. The principal analysis will be the incremental cost per quality adjusted life year gained from the health and social services cost perspective using the EQ-5D-5L to calculate quality adjusted life years (QALYs) as recommended by National Institute for Health and Care Excellence (NICE).[48] A secondary analysis will include the cost per capability adjusted life year (CALY) gained of the antipsychotic reduction strategy compared with maintenance treatment over 24 months using the ICECAP-A, argued to have better face validity.[49] Additional sensitivity analyses will explore the cost-effectiveness of antipsychotic reduction compared with maintenance capturing the quality of life loss associated with antipsychotics side effects using specific disutility values for antipsychotic side effects from the published literature.[50]

The WPAI questionnaire and associated formula[44] will be used to give a monetary value to impact on employment. Health and social care data costs using published sources will be used to calculate costs for the primary health and social care analysis. A secondary analysis will be conducted from a societal perspective to capture the impact on employment, criminal justice, benefits, family and close others.

All costs and outcomes will be discounted in line with NICE guidance at 3.5% per year. Bootstrapping will be used to construct CIs for total mean costs, QALYs and CALYs and to construct cost-effectiveness acceptability curves and cost-effectiveness planes. Missing data will be

handled in the same way as stated by the statistical analysis plan.

## Sample size calculation

The primary outcome is social functioning, measured using the Social Functioning Scale.[31] The literature suggests a minimum difference of between four and five points would be clinically significant, since a difference of four to five points appears to differentiate between patients with good and poor outcomes.[37 51 52]

Using a conventional α of 5% (two-sided) and taking a SD of 8.8 derived from the literature,[53] a sample size of 134 evaluable participants is required to provide 90% power to identify a five point difference, and a sample of 206 is required to identify a four point difference.

The trial also aimed to establish the safety of the antipsychotic reduction strategy by being powered to detect differences in rates of 'severe' relapse (psychiatric hospitalisation), the principal safety measure. We believe that an increased risk of up to 10% would be acceptable to many clinicians and patients if it is balanced by the important outcomes of an improvement in social functioning and reduced side effects. We derived event rates for hospitalisation from those reported in the Leucht *et al*[6] meta-analysis of antipsychotic discontinuation studies.

We conducted a non-inferiority calculation using a 10% margin of difference. Using a non-inferiority boundary of 10% event rates (severe relapse), with an alpha of 0.05, requires a sample size of 372 for 90% power to exclude a difference of 10% between groups. Adding 15% for attrition brings this up to 402. The lower CI on the absolute scale would exclude a difference of 10% in the situation where non-inferiority was achieved. Four hundred and two participants provides 90% power to detect a difference of 3.1 points on the primary efficacy outcome (the SFS), including an allowance of 15% for attrition or unexpected challenges to the assumptions.

It is acknowledged that the non-inferiority approach may be unhelpful in these circumstances, as different people may have substantially different views on what is a reasonable non-inferiority boundary. Further, the assumptions required on issues such as the event rate for the non-inferiority assumption are strong, and modest changes in rates can have a substantial impact on the statistical power of the study making the proposed non-inferiority boundary unhelpful in certain circumstances. Thus the principal aim of the study with regards to severe relapse is to provide a robust estimate of the event rates in each randomised condition, and the difference between them, describing measurement error with appropriate CIs in order to allow individual patients and their clinicians to make an informed decision on treatment options.

## PATIENT AND PUBLIC INVOLVEMENT

Testing this research question is a response to a long-standing demand from the service user and carer community for an alternative to long-term antipsychotic medication. Many service users and carers are extremely concerned about the possible long-term side effects of antipsychotic medication. The study has four co-applicants with a patient or public perspective who attend project management meetings; a carer, a service user, an experienced coordinator and researcher and a staff member of a relevant charity. There is also a LEAP, made up of people with expertise in antipsychotic medication through personal use or as a carer for someone with psychosis. It meets regularly to discuss the progress of the study and to contribute to its development, for example, advising on the antipsychotic reduction strategy. The LEAP will be particularly involved in the qualitative study as part of the process evaluation. They will also be involved in providing feedback on study outputs including a regular newsletter and disseminating research results. Results of the study will be disseminated to participants at their request.

## MONITORING

The trial is monitored by a representative of the Sponsor according to the agreed trial-specific monitoring plan. A Data Safety and Monitoring Board (DSMB) and Programme Steering Committee provide independent oversight of the trial. The DSMB safeguards the interests of trial participants by assessing the safety and efficacy of the interventions during the trial, and monitoring its conduct and it makes recommendations to the steering committee. There will be no formal interim analysis, but the DSMB will continually review all adverse events data. The trial will be stopped if it is judged that there is a substantial increase in serious adverse events that are likely to be related to the intervention.

## SAFETY AND ADVERSE EVENTS

All adverse events are recorded with a description of the event, and are assessed for severity, causality, expectedness and seriousness.

## ETHICS AND DISSEMINATION

The trial was registered with the International Standard Randomised Controlled Trials register on 7 February 2017 and with ClinicalTrials.gov on 15 June 2018. All trial data is handled according to the UK Data Protection Act 1998. Substantial protocol amendments are documented and submitted for ethical and regulatory approval prior to implementation.

Written informed consent is obtained prior to participation in the trial, following a full explanation of the aims, methods, anticipated benefits and hazards of the trial. A formal assessment of each participant's capacity to provide consent is completed first. Consent is an ongoing process and researchers ask participants for verbal consent at each follow-up time point.

Results will be published in peer-reviewed journals and disseminated to the public and the media. All contributors

must meet the 'Authorship Criteria', as recommended by The International Committee of Medical Journal Editors.

## Current trial status

Recruitment started in March 2016 and will continue until 2020.

## DISCUSSION

Better evidence is required about the balance of risks and benefits of a supported programme of antipsychotic reduction and discontinuation under the guidance of a clinician for the many people who have a diagnosis of schizophrenia or recurrent psychoses. The current study will be the first study conducted with people with more than one psychotic episode to employ a gradual and supported method of antipsychotic reduction, to look at a wide range of outcomes giving priority to social functioning and to follow people up for a reasonable duration. Further follow-ups are envisaged after the official end of the trial in order to provide data on longer-term outcomes.

The Dutch first-episode study suggested that long-term outcomes after a period of antipsychotic dose reduction differ from short-term outcomes.[24 54] Although relapse rates are increased initially compared with maintenance treatment, over time these equalise. Social functioning, which was not affected in the short-term, was considerably improved by the time of the 7 year follow-up assessment. Social functioning is a measure of independence and personal efficacy. If the current trial results in improvements in social functioning this could reduce an individual's reliance on services and produce significant economic benefits. From the individual's point of view, better social functioning reflects a more fulfilling and socially integrated life. A strategy that can successfully reduce the use of antipsychotic medication is also likely to be associated with health benefits and improvements in quality of life secondary to the reduction of adverse effects.

The outcomes of the current trial will provide good evidence to inform patients and clinicians about the likely consequences of reducing and discontinuing long-term antipsychotic treatment in a gradual manner in a clinical setting. Many patients currently want to consider this option, but existing data on the range of relevant outcomes is limited. Providing further information will facilitate a more collaborative approach to long-term antipsychotic treatment.

## Author affiliations

[1]Division of Psychiatry, University College London and North East London NHS Foundation Trust, London, UK
[2]Division of Psychiatry, University College London, London, UK
[3]Institute for Clinical Trials and Methodology, University College London, London, UK
[4]Division of Psychiatry, Imperial College London, London, UK
[5]McPin Foundation, London, UK
[6]Research Department of Primary Care and Population Health, University College London, London, UK
[7]Independent consultant, Brentwood, UK
[8]Independent consultant, Brighton, UK
[9]Independent consultant, London, UK
[10]School of Pharmacy, University College London, London, UK
[11]Unit for Social and Community Psychiatry, Queen Mary University of London, London, UK

**Acknowledgements** The authors would like to thank members of Priment for their support for the trial. The RADAR research team also acknowledges the support of the National Institute for Health Research Clinical Research Network and North East London NHS Foundation Trust. The research team thank the RADAR Lived Experience Advisory Panel (LEAP) for their expertise, guidance and support. With thanks also to the independent members of the Programme Steering Committee and the Data and Safety Monitoring Board.

**Contributors** JM developed the original concept for the trial and drafted the protocol. All authors contributed to the trial design and methodology. JM, GL, NF, SJ, TB, NM, VP, RH, LK, RS, KD, RH and SP are grantholders. NF provided statistical expertise. NC and RC provided trial management and updated the protocol. JM drafted the manuscript, and JM and NC edited subsequent drafts. All authors reviewed and commented on drafts of the paper and approved the final manuscript version. The study Sponsor has contributed to the study design, data collection, management, analysis and interpretation of data.

**Funding** This research is sponsored by Priment Clinical Trials Unit, University College London (Sponsor's reference number: 15/0947). This report is independent research funded by the National Institute for Health Research (Programme Grants for Applied Research, Research into Antipsychotic Discontinuation And Reduction (RADAR), RP-PG-0514-20004).

**Disclaimer** The views expressed in this publication are those of the author(s) and not necessarily those of the National Health Service, the National Institute for Health Research (NIHR) or the Department of Health. The NIHR will not have any role during the study execution, analyses or interpretation of data.

**Competing interests** TREB: In the last three years, TREB has been a member of scientific advisory boards for Otsuka/Lundbeck, Newron Pharmaceuticals and Gedeon Richter and received speaker fees from Janssen. GL has acted as an expert witness in cases concerning litigation about antidepressants.

**Patient consent for publication** Not required.

**Ethics approval** London - Brent Research Ethics Committee, 16/LO/1507. Approval letter is uploaded.

**Provenance and peer review** Not commissioned; externally peer reviewed.

**ORCID iDs**
Sonia Johnson http://orcid.org/0000-0002-2219-1384
Nicola Morant http://orcid.org/0000-0003-4022-8133
Rachael Hunter http://orcid.org/0000-0002-7447-8934

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
