## [Reviewer comments · BMJ Open]

ARTICLE DETAILS

TITLE (PROVISIONAL)	A randomised controlled trial of gradual antipsychotic reduction and discontinuation in people with schizophrenia and related disorders: The RADAR trial (Research into Antipsychotic Discontinuation and Reduction) Protocol paper
AUTHORS	Moncrieff, Joanna; Lewis, Glyn; Freemantle, Nick; Johnson, Sonia; Barnes, Thomas; Morant, Nicola; Pinfold, Vanessa; Hunter, Rachael; Kent, Lyn; Smith, Ruth; Darton, Katherine; Horne, Robert; Crellin, Nadia; Cooper, Ruth; Priebe, Stefan

VERSION 1 – REVIEW

REVIEWER	Jimmy Lee Institute of Mental Health, Singapore
REVIEW RETURNED	26-Apr-2019

GENERAL COMMENTS	The authors have proposed to do an important study, to guide antipsychotic maintenance practice in people with schizophrenia and related disorders. I have some concerns and clarifications relating to this study below. 1. My main concern: The team has lumped antipsychotic reduction and discontinuation into a single entity/intervention. In my opinion, the two are not the same. There have been studies on antipsychotic dose reduction, to the minimum dose, which might not have significant differences in clinical outcomes but this is not the same as discontinuation where studies have shown high rates or relapse and adverse outcomes. In the event of a participant who have reduced dose gradually for a duration before discontinuation, the risk of relapse might have commenced from the time antipsychotic doses were discontinued, rather than when the doses were reduced. The analyses should take this point into consideration. Though ideally, there might have been 3 different arms; maintenance, dose reduction, discontinuation. 2. The choice of social functioning as primary outcome also needs some clarification. If the evidence for dose reduction/discontinuation relates to relapses, and that's also the clinical concern, shouldn't the study consider relapse as the primary outcome measure instead. Furthermore, the role of antipsychotic treatment really is about symptom amelioration, which is more direct. Changes in social functioning might not be entirely related to symptoms.
--

	3. In the dose reduction/discontinuation arm, it might be necessary to record rescue medications or dose escalation as secondary outcomes, which might avert the relapse. 4. Would the team consider perceptions or attitudes of clinicians to dose reduction/discontinuation at baseline an important information. How clinicians' perceive or their attitudes towards such an approach might affect the subsequent conduct and validity of the trial. The plan for a qualitative group is an important feature and a strength of this study design.
--	--

REVIEWER	Carsten Hjorthøj Copenhagen Research Center for Mental Health - CORE
REVIEW RETURNED	26-Apr-2019

GENERAL COMMENTS	The authors describe an interesting and important trial to reduce or discontinue antipsychotic medication in patients with schizophrenia and related disorders. Overall, the protocol is well-written. There are some points that should be clarified in the methods section, particularly. These are as follows: The randomization procedure is not described. How is the random sequence generated, and how is allocation concealment achieved. Also, we are not given any details about recruitment. Where do eligible participants come from, how are they screened and invited, how are they informed, etc. Also, where do assessments take place etc. Under "interventions", the reduction schedule is very vaguely described - so vaguely as to make it impossible to know how they are doing it, and thus rendering replications equally impossible. Even if it is not set in stone how it should be done, there must surely be some guidelines or similar that can be described. The list of outcome measures does not distinguish between secondary and exploratory outcome measures. It should. The statistical analysis plan for the primary outcome highlights the use of generalised mixed models, without imputation. Presumably, the authors mean that they use methods that handle missing data through full information maximum likelihood then? The way missing data is handled needs to be explicitly stated. The authors will make "pessimistic assumptions" for missing data - this can be done in nearly infinitely many ways and should be described better. The authors say that they will produce a full statistical analysis plan a priori. I would have preferred for this to be available before they started the trial, but that is too late now. What does "a priori" then mean in this context? Before they start looking at data? And will this SAP be made publically available so that it is possible to check if the others have done what they set out to do.
---

	The sample size calculation mentions both 134, 206, 372 and 402 as possible numbers for sample size. I'm sure the authors have decided on a single number beforehand. We need to know what this is so that we can determine, once the trial is done, if the authors got as many participants as they set out to. This number should then also be highlighted in the abstract. Alpha=0.05, SD 8.8, 90% power, different of five - I get this to require 132 participants, not 134 as the authors describe. The authors use a non-inferiority approach for severe relapse, which seems to make sense. They set alpha at 0.05. Presumably this is a one-sided alpha. I might suggest a one-sided alpha of 0.025 instead. There are no power calculations for secondary outcomes.
--	---

REVIEWER	larry Davidson yale university school of medicine U.S.A.
REVIEW RETURNED	20-May-2019

GENERAL COMMENTS	This is a very timely and important study that is well-designed.
--

REVIEWER	Lex Wunderink GGZ Friesland Netherlands
REVIEW RETURNED	30-May-2019

GENERAL COMMENTS	This is a very important trial, well designed and addressing a very relevant clinical issue. Antipsychotic dose reduction and discontinuation, performed in a flexible, personalized way as implemented in this trial, has not been studied in a multiple episode population before. I recommend publication.
---

VERSION 1 – AUTHOR RESPONSE

Reviewer 1

Comment 1:

1. My main concern: The team has lumped antipsychotic reduction and discontinuation into a single entity/intervention. In my opinion, the two are not the same. There have been studies on antipsychotic dose reduction, to the minimum dose, which might not have significant differences in clinical outcomes but this is not the same as discontinuation where studies have shown high rates of relapse and adverse outcomes. In the event of a participant who has reduced dose gradually for a duration before discontinuation, the risk of relapse might have commenced from the time antipsychotic doses were discontinued, rather than when the doses were reduced. The analyses should take this point into consideration. Though ideally, there might have been 3 different arms; maintenance, dose reduction, discontinuation.

Response 1:

The reviewer expresses concern that we have combined antipsychotic reduction and discontinuation. We agree that there is a distinction between a study that aims to reduce high doses of medication and one that involves discontinuation. The current study involves the evaluation of a pragmatic treatment approach consisting of a process of gradual reduction aiming for discontinuation where possible. The

trial is not a dose reduction trial in the sense of reducing high doses of antipsychotics. The aim for all participants is to discontinue or, if the participant prefers, to reduce to a very low dose that would in most cases be considered as sub-therapeutic. We have specified this in the section on the Intervention and we have adjusted the wording to make this quite clear as follows on P 6:

“The participants are offered the option to discontinue antipsychotic medication completely if the reduction progresses well, or to reduce to a very low dose, defined as the equivalent of 2mg of haloperidol a day or less, which is lower than the minimum recommended therapeutic dose for most antipsychotics.”

We agree with the reviewer that exploring whether there is a difference in rates of relapse between people who discontinue completely and those who only reduce the dose of their antipsychotics will be important.

Comment 2

2. The choice of social functioning as primary outcome also needs some clarification. If the evidence for dose reduction/discontinuation relates to relapses, and that's also the clinical concern, shouldn't the study consider relapse as the primary outcome measure instead. Furthermore, the role of antipsychotic treatment really is about symptom amelioration, which is more direct. Changes in social functioning might not be entirely related to symptoms.

Response 2:

The choice of primary outcome was given careful consideration, but we agree that this could have been more clearly described in the text. Social functioning was chosen above relapse because of the increasing priority given to improving patients' general quality of life, independence and functioning and because some indications from other research suggests that antipsychotics may depress social functioning in the long-term, despite improving symptoms (Wunderink, 2013; Wilks). We have added the following sentence to the Introduction to clarify the basis of our choice:

“Effects on social functioning are particularly important to study because of some evidence that long-term antipsychotic treatment may impair social functioning, despite improving symptoms or reducing relapse in the short-term (Wunderink et al, 2013; Wilks et al, 2016).”

Comment 3:

3. In the dose reduction/discontinuation arm, it might be necessary to record rescue medications or dose escalation as secondary outcomes, which might avert the relapse.

Response to comment 3:

We are recording the use of additional medication, and fluctuations in antipsychotic dose throughout the study in all participants, as suggested, and we will explore the effects of the use of additional medication on outcomes. We have added a sentence specifying that this information is recorded on P 7 as follows:

“All changes in antipsychotic doses and use of other medications is recorded throughout the study.”

Comment 4:

4. Would the team consider perceptions or attitudes of clinicians to dose reduction/discontinuation at baseline an important information. How clinicians' perceive or their attitudes towards such an approach might affect the subsequent conduct and validity of the trial.

Response to comment 4:

We agree that clinician attitudes to antipsychotic reduction and discontinuation and to the study are important. We have not built this into the study as it stands, but we may conduct a retrospective study of participating clinicians in the future.

Final comment:

The plan for a qualitative group is an important feature and a strength of this study design.

Response:

We welcome the comments that the qualitative study is a strength of the study.

Reviewer 2

Comment:

The randomization procedure is not described. How is the random sequence generated, and how is allocation concealment achieved.

Response:

We have added some more detail as follows on P 6 as follows:

“The trial consists of an open, parallel group randomised trial with concealed, individual randomisation. Randomisation is conducted through an independent, internet-based system linked with the database (<https://www.sealedenvelope.com/randomisation/internet/>) with 1:1 allocation.

There are no replacements for participants who drop out or otherwise cannot comply with study procedures.

Participants and clinicians are aware of allocation, but outcome assessors are maintained blind to intervention arm as far as possible. This is achieved by nominating separate blinded and unblinded researchers at each site, and ensuring that blinded researchers are not exposed to discussions or written information that would reveal allocation. Following each assessment point blinded research staff record if they suspect they have guessed arm allocation.”

Comment:

Also, we are not given any details about recruitment. Where do eligible participants come from, how are they screened and invited, how are they informed, etc. Also, where do assessments take place etc.

Response:

A section on recruitment has been added as follows on P 10:

“Participants are recruited from a variety of clinical teams within mental health services across the United Kingdom. Potential participants are identified initially by clinical staff or recruited by advertisements placed in clinical settings. Those who agree are sent further information about the study and then a baseline assessment is arranged after further discussion. Assessments are conducted at the patient’s home or in clinical premises according to patient preference.”

Comment:

Under "interventions", the reduction schedule is very vaguely described - so vaguely as to make it impossible to know how they are doing it, and thus rendering replications equally impossible. Even if it is not set in stone how it should be done, there must surely be some guidelines or similar that can be described.

Response

We have added some more details about the reduction schedules on P 6 as follows:

“The dose is reduced incrementally every one or two months, focusing on one drug at a time where participants are taking more than one antipsychotic. The rate of reduction varies according to baseline dose, with most schedules aiming for discontinuation within 12 months, but some lasting longer where baseline doses are high.”

Comment:

The list of outcome measures does not distinguish between secondary and exploratory outcome measures. It should.

Response:

We did not differentiate between secondary and exploratory outcomes in our protocol, since this is not common practice in trial methodology in the United Kingdom. All the measures listed are secondary outcome measures.

Comment:

The statistical analysis plan for the primary outcome highlights the use of generalised mixed models, without imputation. Presumably, the authors mean that they use methods that handle missing data through full information maximum likelihood then? The way missing data is handled needs to be explicitly stated.

Response

We are surprised that the referee has come to the conclusion above. We have stated that 'The principal analysis will be undertaken using the intention to treat population, including all available data, without imputation.' We also spell out how we intend to parameterise the model, and indicate the use of a detailed statistical analysis plan. We have therefore made no changes to the manuscript on this point.

Comment:

The authors will make "pessimistic assumptions" for missing data - this can be done in nearly infinitely many ways and should be described better.

Response:

We state in the methods section that we are assessing the effect of missingness using a threshold analysis and it is thus quite precise.

Comment:

The authors say that they will produce a full statistical analysis plan a priori. I would have preferred for this to be available before they started the trial, but that is too late now. What does "a priori" then mean in this context? Before they start looking at data? And will this SAP be made publically available so that it is possible to check if the others have done what they set out to do.

Response:

We have added further specification on P 12 that the SAP will be completed before database lock.

We are happy to make the SAP publicly available:

"A full statistical analysis plan will be completed prior to database lock, describing planned subgroup analyses and with detailed description of the statistical processes to be used."

Comment:

The sample size calculation mentions both 134, 206, 372 and 402 as possible numbers for sample size. I'm sure the authors have decided on a single number beforehand. We need to know what this is so that we can determine, once the trial is done, if the authors got as many participants as they set out to. This number should then also be highlighted in the abstract.

Response:

There are different power calculations, because we calculated sample sizes for the primary outcome (social functioning) at different effects sizes, and also for difference in relapse rates (a secondary outcome). We have added more information on this to the Abstract as follows:

"Secondary outcomes include severe relapse (admission to hospital) and the study is also intended to detect a minimum 10% difference in severe relapse, which requires 402 participants, assuming a 15% loss to follow-up."

Comment:

Alpha=0.05, SD 8.8, 90% power, different of five - I get this to require 132 participants, not 134 as the authors describe.

Response: On our analyses it requires 132 df, which might be the confusion? It requires 134 subjects.

Comment:

The authors use a non-inferiority approach for severe relapse, which seems to make sense. They set alpha at 0.05. Presumably this is a one-sided alpha. I might suggest a one-sided alpha of 0.025 instead.

Response:

The alpha level used is actually two sided at 5% which is analogous to one sided at 0.025 for the non inferiority comparison. However we are aiming to estimate the effects described rather than embracing hypothesis testing, and so we are interested in both confidence intervals not just the lower one.

Comment:

There are no power calculations for secondary outcomes.

Response:

We defined the desired statistical power at the outset and then conducted a sample size calculation based on the effect size that we assumed as important for the primary outcome and, for safety reasons, for the secondary outcome of severe relapse. There is no pre-specified alpha spend for other secondary outcomes and all are considered nominal.

VERSION 2 – REVIEW

REVIEWER	Carsten Hjorthøj Copenhagen University Hospital, Mental Health Center Copenhagen
REVIEW RETURNED	29-Jul-2019

GENERAL COMMENTS	In response to my comment: "Under "interventions", the reduction schedule is very vaguely described - so vaguely as to make it impossible to know how they are doing it, and thus rendering replications equally impossible. Even if it is not set in stone how it should be done, there must surely be some guidelines or similar that can be described.", the authors provide the following response: We have added some more details about the reduction schedules on P 6 as follows: "The dose is reduced incrementally every one or two months, focusing on one drug at a time where participants are taking more than one antipsychotic. The rate of reduction varies according to baseline dose, with most schedules aiming for discontinuation within 12 months, but some lasting longer where baseline doses are high." However, it is still rather vaguely described, but perhaps this is intentional? For example, is a specific reduction plan made for each participant, with some clinical judgement involved? If so, that is fine but I would like this to be stated explicitly. If an algorithm exists detailing the exact reduction intended at different times, perhaps with different algorithms for different drugs, then I would very much like, as a reader, to see this / these algorithms. In response to my comment: "The statistical analysis plan for the primary outcome highlights the use of generalised mixed models, without imputation. Presumably, the authors mean that they use methods that handle missing data through full information maximum likelihood then? The way missing data is handled needs to be explicitly stated.", the authors provide the following response: "We are surprised that the referee has come to the conclusion above. We have stated that 'The principal analysis will be undertaken using the intention to treat population, including all
--

	available data, without imputation.’ We also spell out how we intend to parameterise the model, and indicate the use of a detailed statistical analysis plan. We have therefore made no changes to the manuscript on this point.”. I think I may have phrased my point incorrectly. I agree that mixed models are generally an adequate method for analysis of longitudinal data, even in the presence of missing information. My point was simply to highlight that this method handles missing data through full information maximum likelihood, and that this is why it is fine to perform this analysis even without imputations. As such, I still think the text could be slightly amended in this regard, but only in terms of helping the reader to understand that the authors are actually using state-of-the-art methodology.
--	--

VERSION 2 – AUTHOR RESPONSE

Comment:

Please leave your comments for the authors below In response to my comment: "Under "interventions", the reduction schedule is very vaguely described - so vaguely as to make it impossible to know how they are doing it, and thus rendering replications equally impossible. Even if it is not set in stone how it should be done, there must surely be some guidelines or similar that can be described.", the authors provide the following response: We have added some more details about the reduction schedules on P 6 as follows: “The dose is reduced incrementally every one or two months, focusing on one drug at a time where participants are taking more than one antipsychotic. The rate of reduction varies according to baseline dose, with most schedules aiming for discontinuation within 12 months, but some lasting longer where baseline doses are high.” However, it is still rather vaguely described, but perhaps this is intentional? For example, is a specific reduction plan made for each participant, with some clinical judgement involved? If so, that is fine but I would like this to be stated explicitly. If an algorithm exists detailing the exact reduction intended at different times, perhaps with different algorithms for different drugs, then I would very much like, as a reader, to see this / these algorithms.

Response:

We are grateful to the referee for asking for further clarification. No algorithms were used. Schedules were drawn up for each participant based on clinical judgement. We have amended the text to reflect this as follows: “an individualised reduction schedule is devised by the research team for each participant based on clinical judgement and adjusted according to the participant’s initial antipsychotic regimen.”

Comment:

In response to my comment: "The statistical analysis plan for the primary outcome highlights the use of generalised mixed models, without imputation. Presumably, the authors mean that they use methods that handle missing data through full information maximum likelihood then? The way missing data is handled needs to be explicitly stated.", the authors provide the following response: "We are surprised that the referee has come to the conclusion above. We have stated that ‘The principal analysis will be undertaken using the intention to treat population, including all available data, without imputation.’ We also spell out how we intend to parameterise the model, and indicate the use of a detailed statistical analysis plan. We have therefore made no changes to the manuscript on this point.”. I think I may have phrased my point incorrectly. I agree that mixed models are generally an adequate method for analysis of longitudinal data, even in the presence of missing information. My

point was simply to highlight that this method handles missing data through full information maximum likelihood, and that this is why it is fine to perform this analysis even without imputations. As such, I still think the text could be slightly amended in this regard, but only in terms of helping the reader to understand that the authors are actually using state-of-the-art methodology.

Response:

We understand the point the referee is making, but we are not convinced that changing the text is required or will help clarify the analysis. The terminology that the referee employs is specific to Stata and will not be understood by ordinary readers with a less specific understand of our methods.

VERSION 3 – REVIEW

REVIEWER	Carsten Hjorthøj Copenhagen University Hospital, Mental Health Center Copenhagen
REVIEW RETURNED	26-Aug-2019
GENERAL COMMENTS	I thank the authors for their revision based on my first comment. While I do not completely agree with the authors' rationale for not making any changes based on my second point, it is not important enough that I will argue the point any further. Consequently, I am happy to recommend that the manuscript be accepted for publication in its current form. I wish the authors the best of luck in conducting their trial.